# The Participation of Trans Women in Competitive Fencing and Implications on Fairness: A Physiological Perspective Narrative Review

**DOI:** 10.3390/sports11070133

**Published:** 2023-07-17

**Authors:** Victoria Tidmas, Clare Halsted, Mary Cohen, Lindsay Bottoms

**Affiliations:** 1Centre for Research in Psychology and Sport Sciences, University of Hertfordshire, Hatfield AL10 9EU, UK; v.tidmas@herts.ac.uk; 2British Fencing, London W4 5HT, UK; clarehalsted@blueyonder.co.uk (C.H.); marybeatrixcohen@gmail.com (M.C.)

**Keywords:** trans women, elite fencing, fair, gender differences, male physiological advantage

## Abstract

Debate has surrounded whether the participation of trans women in female sporting categories is fair, specifically the retained male physiological advantage due to increased testosterone compared to cisgender females. Recently, individual sporting organisations have been investigating and assessing policies regarding trans women athlete participation in female categories, resulting in several banning participation. This review aims to discuss the scientific evidence and provide appropriate guidance for the inclusion of trans women in elite competitive female fencing categories. Fencing is an intermittent sport, where competitions can span 1 to 3 days. The lunge is the most common movement used to attack opponents, where a successful hit relies on the speed of the action. Male puberty induced increased circulating testosterone promotes a greater stature, cardiovascular function, muscle mass, and strength compared to cisgender females, culminating in a ~12–40% sport performance advantage. Elite cisgender male fencers perform significantly higher, ~17–30%, jump heights and leg power measures compared to elite cisgender female fencers, resulting in faster lunges. Trans women receiving androgen-suppression therapy for 12 months showed significant reductions in strength, lean body mass, and muscle surface area, but even after 36 months, the measurements of these three indices remained above those for cisgender females. Previous male muscle mass and strength can be retained through continuation of resistance training. The literature reviewed shows that there is a retained physiological advantage for trans women who have undergone male puberty when participating in the elite competitive female fencing category. A proposed solution of an open or third gender category for elite fencing competition promotes fair competition, while allowing trans women to compete in their chosen sport.

## 1. Introduction

Most competitive sports, including elite fencing, are segregated into male and female sex categories, due to a large and persistent performance gap, whereby males outperform female athletes [1,2,3]. Recently, reports have shown an increasing number of people identifying as transgender [4,5] and transgender athletes in elite and recreational sports [6,7]. Transgender individuals, abbreviated to “trans” throughout this review, have a gender identity that conflicts with their sex recorded at birth [8]. Trans men athletes (female transitioned to male), who use cross-sex hormones, such as testosterone, to medically transition, are not seen to possess any athletic advantage when competing in male sporting categories [9]. Considerably more attention has focused on trans women (male transitioned to female) athletes, due to the retention of male physiological performance advantages, mainly testosterone driven, compared to cisgender female athletes [8,9]. Individuals whose gender identity is the same as their sex allocated at birth will be described as “cis”, short for cisgender, throughout this review. Currently, one of the most prevalent and controversial topics is the development of eligibility criteria for the inclusion of transgender individuals in sports, especially trans women, with an emphasis on how to maintain fairness in competition, while also including transgender athletes in the category that conforms with their gender identity [8,9,10,11,12,13,14,15,16]. 

### 1.1. Transgender Inclusion Policies and Guidelines

Previously, the International Olympic Committee (IOC) guidelines [17] permitted trans women to compete in female-only category sporting events if they had received gender affirmation surgery. This requirement was removed, so that the 2015 guidelines stated that trans women athletes need to demonstrate at least 12 months of total testosterone levels <10 nmol/L and need to have identified as female for at least 4 years to qualify [18]. In 2021, the IOC changed its approach and published its Framework on Fairness, Inclusion and Non-Discrimination on the Basis of Gender Identity and Sex Variations [16], which was updated in 2023 [19]. This framework has been extremely influential for other sport organizations’ policy development, as it now recommends that individual sports should investigate whether the inclusion of trans women athletes in female categories is safe and fair when considering changes to policies for participation within the sport. As a result, various sporting governing bodies have established changes to their sporting discipline’s eligibility criteria, which are being constantly reviewed and updated. For example, the world governing body for cycling, the Union Cycliste Internationale (UCI), in March 2022, [20] and World Triathlon, in August 2022 [21], enforced stricter rules for trans women athletes to be able to compete in female categories, requiring testosterone levels of ≤2.5 nmol/L for an extended 24-month period. World Rugby, in March 2021 [22], the international swimming federation F.I.N.A, as of June 2022 [23], and World Athletics, as of 31 March 2023 [24], have banned trans women from participating in female categories if the athlete has experienced any part of male puberty, to maintain sporting integrity and athlete safety. Other participation options are available, such as an open category for those who identify differently to their sex at birth or for those with sex variations [12].

### 1.2. Needs Analysis of Fencing

#### 1.2.1. Physiological Demands of Fencing

Fencing is an intermittent sport with three weapon categories: sabre, épée, and foil. During a fight, fencers undertake multiple explosive actions (particularly lunging movements) to outscore their opponent. Typical work to rest ratios of 1:1.2 are observed for épée [25], 1:3 for foil [26], and 1:6.5 for sabre [27], whereby fencers will work for an average of 10, 5, and 2.5 s per action, respectively. Therefore, there are distinct differences between the weapon categories; sabre is more explosive, whereas épée has a greater submaximal component. Differences between weapons are primarily due to the specific rules of the sport, whereby foil and sabre observe a priority system [26], where starting the first attack is important for scoring. Épée has no priority system, and the tactics therefore produce longer phases during a bout. International fencing competitions last between 1 and 3 days. Although each competition day can last up to 12 h, of which the actual fencing time has been estimated between 5 and 15% of the competition day [26]. Fencing competitions run in a similar format to a major football tournament, comprising an initial phase of pool fights in a round robin stage (first to 5 points or maximum of 3 min plus an additional minute if scores are even), which then ranks fencers for subsequent knockout rounds of direct elimination (DE) fights (first to 15 points, timed 3 × 3 min bouts with 1 min rests). There are frequent rest periods between fights, which can last from 5 min to several hours.

Within competitive fencing, there is an emphasis on the phosphocreatine and aerobic energy systems, consequently demonstrating low concentrations of blood lactate [28]. The cardiorespiratory demands during fencing competitions need to be considered as an important training component. For example, competitive fencers typically exhibit maximal oxygen uptake values within the range of ~50–60 mL.kg^−1^.min^−1^ for men [25] and ~40–50 mL.kg^−1^.min^−1^ for women [29]. The mean estimated oxygen cost of both foil and épée competition has been found to be between 56 and 74% maximal oxygen uptake [30] with average heart rate (HR) values as high as 96% of maximal heart rate (HRmax [28]).

#### 1.2.2. Lunge

In fencing, the lunge is the most common movement used to attack opponents, with ~140 attacks per elimination round, necessitating repeatable and accurate movement for fencing success [31,32]. A lunge enables fencers to strike the opponent, while maintaining an out-of-range position. Large concentric forces exerted by the rear leg propel the fencer’s body anteriorly, while maintaining rear foot ground contact [33]. Most upper-body combat sports rely on explosive lower body forces to produce forceful strikes [34]; in fencing, striking as quickly as possible is the aim, therefore attacking manoeuvres rely on the speed of actions [31,35,36,37,38]. 

Kinematic analysis of a successful fencing lunge shows a set sequential activation of muscles: first, dominant arm extensor elbow muscles; followed by hip, knee, and ankle muscles [26,33,39,40,41,42]. The sequential nature of both upper and lower body parts necessitates exceedingly well coordinated decisive actions during all fencing manoeuvres [43,44].

Strength and power are both determinants of lunging and changing direction, which are essential for fencing performance [32,45]. Research has shown that better squat jumps, counter movement jumps, and reactive strength index results were significantly correlated with improved performance in lunge times and shuttle test scores [46]. The best predictor of both lunge time and shuttle test score was the squat jump, with all other lower-body power tests also proving significant. Suggesting that the stretch–shortening cycle mechanism, concentric explosive strength, and maximum strength are particularly important for fencing-related movements patterns and performance [26,31,35,43,47]. 

### 1.3. Methodological Considerations and Aims 

The recent changes to sport’s governing body transgender policies have been based on the limited research observing the clinical outcomes of transgender participants experiencing medical transitions (hormone replacement/suppression treatment), not elite athletes [12,17,48]. These studies often varied in their quality, design, methodology, and sample size. A systematic review of 24 studies investigating non-athletic trans women found that the sample size varied but was usually small (between 11 and 230 participants); observed participants of various ages groups, with some including adolescents; and that transition medical treatments were individualised, substantially affecting participants hormones levels and physiological characteristics [49]. No relevant research was found observing the effect of transition medical treatments taken prior to puberty. More research is needed to observe the effect of the longer durations of hormone treatments on elite transgender athletes and to identify additional biomarkers alongside using serum testosterone for eligibility criteria [50]. However, there is opinion that the general findings from non-athletic trans women would be applicable if not exacerbated for trained trans women [18,49,50,51]. The authors acknowledge there is currently no research investigating elite trained transgender athletes’ participation in fencing. Hence, this review discusses research from comparable groups of individuals, such as the physiological differences between cis males and cis females, differences between cis male and cis female fencing performance parameters, those with genetic deviations, and testosterone suppressed cis males, to provide the best possible overview of physiological evidence to advise the appropriate parties. 

Therefore, the aim of this review was to present the scientific evidence relating to the potentially unfair physiological advantage trans women may sustain over cis female athletes in fencing. Of specific relevance are key physiological performance indicators and determinants of fencing success. The evidence discussed throughout this review may aid in producing guidelines for trans women inclusion within elite competitive female fencing categories and provide appropriate advice for fencing organisations. However, policy updates should also consider other aspects not covered within this review, such as the social or political implications of policy changes to ensure fairness, safety, and inclusivity [12,16,48].

## 2. Physiological Cisgender Male vs. Cisgender Female Differences

Although the sex difference in world record performance slowly declined until the 1990s [52,53], the gap has remained steady over recent decades [1,2], with cis males outperforming cis females by 8–12% [3]. Hence, most sports are sex segregated, to negate the biological advantage of cis males over cis females [53,54]. The main reason for this difference in performance is testosterone [51,54]. Until puberty, the quantity of circulating hormones for young cis males and cis females is essentially the same [54]. The onset of male puberty triggers a hormonal cascade and consistently results in 10–15-times greater circulating testosterone levels (7.3–12.6 nmol/L) for cis males compared to cis females (0–1.7 nmol/L) [54,55,56,57]. Ultimately, post-pubescent cis males possess several significant anthropometric and physiological differences (Table 1) compared to cis females, which correspond to a post-pubescent divergence in athletic performance between males and females [55,58], which this review will focus on. However, there are other factors such as, glucose utilisation [59,60], body fat distribution [61], lipid metabolism [60], energy balance [60], oestrogen levels [54], and skeletal muscle fibre type [62], all of which can affect athletic performance and are all affected by testosterone levels.

Increased circulating testosterone levels in cis males drive increased lung volumes, resulting in greater aerobic capacity [54,63]. In addition, augmented haemoglobin concentrations, ~12% above cis female concentrations, and increased heart size promote an increased cardiovascular function in cis males over cis females [64,65] (Table 1). Larger and longer skeletal bones for cis males is another consequence of the higher testosterone levels compared to cis females, as most cis females experience puberty earlier than cis males, which results in earlier oestrogen-dependent terminated bone growth, resulting in cis females being on average ~7–8% shorter than post-pubescent cis males, with shorter, weaker, and less dense bones [54,66]. Being of taller stature with longer limbs is an advantage for cis males, as these provide increased fulcrums, producing greater leverage and ultimately greater muscular limb power in sports involving throwing, jumping, and explosive power [54]. A larger skeleton, for example broader shoulders, provides greater surface area for increased muscle attachment and growth [8,55,67,68], resulting in considerably higher muscle mass percentage [54,67,69] (Table 1). As such, post-pubescent young cis males possess on average >12 kg more skeletal muscle mass compared to age- and weight-matched cis females [69]. The greater muscle mass of cis males results in significant differences in muscle strength. Cis females have been shown to possess up to 44% less upper-body strength compared to cis males, providing cis males an advantage in upper-body strength dominant sports, such as kayaking, rowing [3], and skiing [70] (Table 2). Decreased trunk and lower body strength have also been shown for cis females compared to cis males, by up to 64% and 72%, respectively [69,70,71]. In sports such as weightlifting where weight categories are different between male and female athletes, cis males of similar weight to cis females (in the earlier 69 kg and newer 55 kg common category) have lifted ~30% greater weight; however, as bodyweight categories increase, so does the performance difference, with top/open weight males lifting ~40% greater weight than top/open females [18]. The performance difference is greater when assessing powerlifting; the World Open Classic Records show that the open male records for squat, bench press, and deadlift total weight are ~65% greater than the female equivalents [18]. Even females that are 60% heavier than males do not out lift lighter males (1998–2018 69 Kg Male combined record = 359 Kg vs. all-time top/open category female combined record = 348 Kg) [18]. 

In terms of lower body strength and power performance, assessments via various ballistic jump heights, for example squat jumps and counter movement jumps, are good indicators for both adult and young individuals [72]. A significantly greater increase in jump performance and leg power was found for cis boys from 11 to 18 years compared to cis girls [58,73,74], which conforms with the onset of puberty. A significantly greater body fat percentage for cis girls was found for all the age groups assessed between 11 and 16 years, while lean body mass was significantly increased for cis boys from 13 to 16 years but not cis girls [75]. Leg muscle volume and leg length were both significantly increased for cis boys from 14 years compared to cis girls, corresponding to significantly increased results for all parameters of jump performance for cis boys from 13 to 16 years, while all results were similar between cis boys and cis girls at age 11 years [75]. Therefore, post-pubescent cis males possess a resultant variable performance advantage over cis females in elite sport, ranging from ~12% for swimming and rowing events to ~30% for combat sports [51] (Table 2). The cumulative performance advantage for cis males over cis females, for example in running where cis males are ~12% faster than cis females (Table 2), corresponds to ~10,000 adult male 100 m sprinters and those of junior male record holders whose personal best times are faster than the 2016 Olympic female 100 m champion and female record holder performances for other distances [18].

## 3. Cisgender Male vs. Cisgender Female Fencing Performance Differences

Multiple studies have assessed the gender differences between cis male and cis female fencers for various determinants linked to successful fencing performance [27,37,76]. Body size and mass are key factors of fencing performance success and are especially important throughout the early years of training [37]. Cis male anthropometric variables were significantly different than cis female fencers, resulting in cis male fencers being taller, longer limbed, and heavier than cis female fencers (Table 3). 

Not surprisingly, cis male fencers significantly outperformed similar-aged cis female fencers in all performance measures [37,77]. Cis male fencers showed a significant increase varying between 17 and 30% over cis female fencers in all jump height and leg power parameters [37] (Figure 1). The counter movement jump (CMJ) was the best indicator of step-lunge success, although technique, aerobic capacity, and tactics play substantial roles and should not be discounted [77]. Results for cis male fencers for CMJ and counter movement jump power (CMJP) showed the greatest difference between cis males and cis females, with cis males outperforming cis females by +28% and +30% respectively (Figure 1). Cis males were significantly faster in all specific fencing offensive kinetic patterns than similar aged cis female fencers [77]. Another recent study found that cis male fencers were faster than cis female fencers during a step-lunge (2.68 ± 0.68 m/s vs. 3.55 ± 0.69 m/s, respectively) and lunge (2.99 ± 0.64 m/s vs. 3.98 ± 0.92 m/s, respectively) [76]. Therefore, post-pubescent cis male fencers have a significant performance advantage compared to cis female fencers. This is reflected in the fact that cis male fencers, sabre fencers especially, have been found to move faster, attack opponents more frequently, and significantly differ in work to rest ratio compared to similarly matched cis female fencers [27]. 

Age also impacts on leg power performance (Figure 2), where older fencers (>20 years) produced significantly higher values for long jump (LJ), squat jump (SJ), CMJ, drop jump (DJ), and reactive strength index (RSI) compared to younger age groups (14–17 and 18–20 years) [37]. More recently, Ntai et al. [77] identified that the age of well-trained fencers also significantly affected specific fencing offensive kinetic patterns, such as lunge velocity (L), step-velocity (SL), and step backward step-lunge (SBSL), where older fencer groups (>20 years and 18–20 years) were faster than younger fencers (14–17 years). This highlights, once again, that fencing-specific adaptations are promoted during growth and maturation phases.

## 4. Testosterone and Females

There are some athletes with rare genetic variations participating within female sports [13,14,15,78]. After unexpectedly and dominantly winning the African Junior Championship 2009 women’s 800 m and the World Championships 2009, the South African athlete Caster Semenya received a lot of scrutiny when a gender test was leaked revealing naturally high levels of testosterone due to internal testes but also an androgen enzyme deficiency [8,13]. There are various intersex conditions, although few promote a female performance advantage [78]. María José Martínez-Patiño, a young Spanish woman hurdler, was disqualified by the Royal Spanish Athletics Federation after her chromosomal constitution was found to not be female. She was later diagnosed with complete androgen insensitivity syndrome (AIS), in which female phenotypic development occurs in the presence of a Y chromosome, due to insensitivity to testicular androgens [15]. It has been estimated that intersex conditions occur in about 1 in 5500 people [79]. However, in athletes this can be much higher, with an estimate of 1 in 500–600 cis female athletes with intersex conditions such as AIS who present with XY chromosomes [80]. Ferguson-Smith [81] found that 1 in 421 cis female athletes over five Olympic games presented with a Y chromosome. Many of these mutations can cause an increased or decreased sensitivity to androgens such as testosterone and lead to performance benefits or deficits when compared to the individuals’ chromosomal gender.

For example, non-athletic hyperandrogenic cis females with slightly increased circulating testosterone levels due to polycystic ovary syndrome (PCOS) were found to possess increased muscle mass relevant to the degree of increased circulating testosterone levels [82,83]. Hyperandrogenism has been shown to be common in elite cis female athletes [84,85,86]; however, it is commonly undiagnosed [87]. This increase in testosterone and muscle mass provides an ergogenic advantage for these athletes [86], which reflects the effects of graded circulating testosterone in males and females and the subsequent difference in explosive power performance [88]. Specifically, for elite cis female athletes, those with significantly higher endogenous testosterone, and who were not intersex, correlated with better performance in certain athletic events (400–800 m running, hammer throwing, and pole vault) [89]. Even within the normal female range of endogenous testosterone, those within the highest quartile still performed between 1.8 and 4.5% better in the events mentioned compared to those in the lowest quartile [89]. A study of 106 Swedish Olympic cis female athletes compared to age- and weight-matched sedentary cis females showed that increased endogenous androgens (testosterone) promoted an anabolic body composition and improved muscle performance [84]. This study found that the athletes possessed greater bone and muscle mass, whereby strength tests correlated strongly with both total and isolated leg muscle mass. Additionally, both increased muscle mass and strength results for the athletes correlated with increased levels of androgens (testosterone) and androgen precursors (dehydroepiandrosterone and androstenedione) [54,84,90,91,92].

Trans men (female to male) receiving testosterone treatment over the course of the first 12 months of treatment have shown significant increases in total body lean muscle mass (+10.4%) and strength [93]. Previously, trans men who achieved circulating testosterone levels of 31 nmol/L (mean value), equivalent to adult cis male levels, showed a 19.2% increase in thigh muscle mass [94]. The administration of testosterone doses equal to those of adult cis males to trans men increased total body muscle size (6.5%) and limb muscle size (16.6%), as well as increasing grip strength by 18% compared to age-matched untreated cis females [93]. A recent study comparing 19 cisgender and 19 trans men participants paired by age and body mass index (BMI) found that, after 24 months of testosterone use, trans men’s testosterone levels were similar to cis men’s values (638 vs. 685 ng/dl; *p* = 0.863, respectively); however, the total body bone mineral density (TBBMD) (1.208 ± 0.132 vs. 1.271 ± 0.081 g/cm^2^, *p* = 0.008), muscle mass (44.09 ± 6.27 vs. 55.71 ± 7.28 kg, *p* < 0.001), and muscle strength (28.82 ± 5.42 vs. 40.34 ± 8.03 kg, *p* < 0.001) for trans men remained less than the values for cis men [95]. Therefore, trans men appear to be at a disadvantage compared to cis males as increases in muscle mass and strength do not equal the percentage advantages for cis males. Additionally, post-female-puberty anthropometric factors cannot be influenced by testosterone supplementation, such as height, limb length, and skeletal size and structure. Therefore, there is less controversy around the inclusion of trans men in the male category in competitive sporting events. Introducing a third gender category in elite sport has been proposed as a solution that would allow transgender individuals, those born with ambiguous sexual anatomy, and those who identify as non-binary to compete in their chosen sporting event [54].

## 5. Testosterone Suppression and Sporting Performance

More elite sports are excluding trans women from participating in female categories, due to concern over the effectiveness of any period of testosterone suppression in eliminating the post-puberty male performance advantage reviewed above. Common testosterone-suppression treatments involve administration of the major female hormone oestradiol, which increases breast tissue development and reduces body hair [8,96]. However, many testosterone-suppressed trans women are still competing with testosterone levels 5-times greater than the upper range exhibited by healthy, premenopausal elite cis female athletes, 0–1.7 nmol/L [54]. Although testosterone is suppressed compared to cis males, it has been shown that trans women retain some of the advantages from their previous male physiology, irrespective of the duration of hormone therapy treatments [8,49]. By contrast, undergoing full gender affirmation surgery with removal of testes has been shown to reduce testosterone levels to castration levels, <1 nmol/L [97]. Androgen suppression therapy has no effect on pre-transition male physiological adaptations experienced post-puberty. Trans women who transition after completing male puberty will therefore possess greater lung volume, heart size, and bone structure, providing advantages involving increased maximal oxygen uptake and stroke volume, as well as joint biomechanics [93,98,99]. However, androgen-suppression therapy affects haemoglobin (Hgb)/haematocrit (HCT) levels [99], where 4 months of hormone therapy resulted in levels in trans women equivalent to cis females, potentially reducing the aerobic capacity of trans women compared to their previous male performance [49]. Trans women experiencing androgen suppression therapy for 2 years showed no signs of bone density reduction, aiding in preventing injury or trauma while promoting recovery from strenuous training and competitions [100]. The addition of oestradiol from hormone therapy can also promote increased muscle mass gains via increased activation of oestrogen receptor-β [101,102,103].

Significant reductions in strength, lean body mass, and muscle surface area were also found after 12 months of androgen-suppression therapy [49]. It should be noted that, even after 36 months of hormone therapy, measurements in trans women of strength, lean body mass, and muscle surface area remained above cis female values [49]. Reducing testosterone to ≤10 nmol/L can result in a reduction of muscle mass; however, it has been shown that young healthy cis males with low testosterone levels equivalent to ~8.8 nmol/L for 20 weeks (normal cis male testosterone level 7.3–12.6 nmol/L) showed no significant loss in muscle mass or subsequent power [104]. 

The administration of graded increasing testosterone doses in fully testosterone-suppressed cis males (~castration levels <1 nmol/L) confirmed a dose-dependent relationship with increasing dosage and increased muscle mass and strength [104,105]. Finkelstein and colleagues [105] also included a placebo dose, which produced a serum testosterone level of 0.7 nmol/L, which is within the normal cis female range and typical for castrated males, while the lowest testosterone dose produced serum testosterone levels equal to 6.9 nmol/L, equivalent to cis male mid-puberty and exceeding normal cis female levels. Significantly, the increase from the placebo (normal cis female testosterone level) to the lowest testosterone dose (supraphysiological cis female concentrations) resulted in increases of total body lean muscle mass, thigh muscle surface area, and leg press strength of 2.3%, 3.0%, and 5.5%, respectively, providing strong evidence that even small differences in circulating testosterone play a large, if not exclusive, role in the sex difference in muscle mass between cis males and cis females. 

Of interest, and potentially comparable to testosterone-suppressed trans women, is androgen deprivation therapy (ADT), a standard treatment for males suffering with and recovering from prostate cancer, as testosterone in particular drives late-stage prostate cancer growth [106]. ADT is known to produce adverse effects, such as decreased libido, increased abdominal fat percentage, and reduction in muscle mass and tone [106,107]. However, the advocation of increased or continued aerobic and/or resistance exercise during and post ADT treatment periods has been shown to negate losses in muscle mass and strength, improve body composition, functional performance, fatigue, and potentially decrease prostate cancer risk and development [108]. Winters-Stone et al. [109] found 1 year of resistance exercise reduced disability and improved physical functioning in prostate cancer survivors on ADT, with results of increased bench press strength and maximal leg strength. The combination of both aerobic and resistance exercise for 3 months at the start of ADT treatment was shown to maintain lean muscle mass and prevent excess fat gain in prostate cancer patients [110]. Resistance training over a period of 6 months promoted increased aerobic fitness and reduction in body fat percentages for prostate cancer patients [111,112]. A meta-analysis by Keilani et al. (2017) showed that prostate cancer patients performing resistance exercise not only had an improved lean body mass (95% CI [0.15–1.84] %; *p* = 0.028) but also significantly improved upper (95% CI [2.52–7.97] kg; *p* < 0.001) and lower body (95% CI [10.51–45.88] kg; *p* = 0.008) muscular strength [108]. While Chen et al. [113] found that seven randomized controlled trials found no significant mean differences in lean muscle mass after resistance exercise for prostate cancer patients, there was still a significant increase in chest (3.15 kg, 95% CI: 2.46, 3.83; *p* < 0.001) and leg press (27.46 kg, 95% CI: 15.05, 39.87; *p* < 0.001) muscle strength.

Strength training and other ergogenic aids promote the retention of muscle mass, even in reduced testosterone trans women and cis males [49]. The phenomenon of muscle memory could be advantageous for trans women athletes, as this enables easily and quickly regaining previous male muscle mass and strength via myonuclei retention, even after significant muscle mass loss, extended periods of inactivity, and in later life [49,114,115]. Training and anabolic steroids increase the number of myonuclei [49,116]. However, myonuclei numbers do not decrease during phases of detraining [115]. Hence, androgen suppression does not fully reverse pre-transition testosterone strength advantages, and trans women can offset any muscle mass losses via continuing strength training, therefore maintaining a significant performance advantage over cis females [49].

## 6. Opinions to Be Considered 

Young and adult cis females experience various barriers to participation in sport, such as social and cultural barriers including lack of time, childcare demands, less funding than males, access to facilities, and fears over personal safety [9]. These often culminate in cis females being less likely to participate in sport than cis males [117] and less represented in elite sport [118]. Consultation of all groups involved, including cis female athletes, is needed, to establish fair and inclusive policy decisions. The new IOC Framework [16,19] has received some criticism, suggesting that the human rights of transgender athletes have been placed above those of cis female athletes [119], even though neither trans women nor cis female athletes is the most dominant group in terms of participation rates and managerial roles in many sports [120]. Cis female Olympians have expressed frustration that the IOC did not consult cis female athletes prior to the policy changes, suggesting to these athletes that the inclusion of trans athletes was prioritised over fair competition [119]. A joint position statement on the behalf of multiple international sport medicine organisations also criticised the IOC framework for disregarding the scientific evidence of the retained physiological performance advantages for trans women athletes and focusing on a mainly human rights perspective [121]. It was also suggested that the IOC places an increased burden on specific sporting governing bodies who are potentially underprepared, underfunded, and lacking the capacity to create and enforce policies meeting the IOC framework [121].

## 7. Open or Third Gender Category 

Approximately 50% of transgender individuals have reported negative experiences including transphobic bullying and harassment from fans, peers, and coaches [122]. Therefore, the updated IOC framework [16] was well received by some LGBT+ community advocacy groups [12,123] for promoting inclusivity and requiring that each governing body assess policy changes specific to each sport, where most organisations have ultimately made decisions on athlete eligibility based on a case-by-case nature [48]. To widen the participation of elite transgender athletes in sports, several approaches have been suggested, such as an open category [23] in addition to a separate cis female category that excludes any individual who has experienced any part of male puberty [124,125]. This is the case for British Triathlon, where in competitions there is separate female and open categories, but no category restrictions for recreational athletes [126]. Additionally, there is the option of a generalised third category available to any competitor. Consideration must be given when enforcing separate groups, to avoid alienating transgender individuals and others who may be affected. There is a link between regulations enforced for elite sport participation and impacts on participation at a grassroot level, especially for young people [127,128]. Separate groups may end up consisting of other groups of athletes excluded from the cis female athletic category, for example, those with intersex variations, culminating in a wide variety of athletes within this open/third group, making fairness harder to ensure within this category. 

Logistically, the number of elite athletes meeting the exclusion criteria and warranting a separate sporting competition category needs to ensure meaningful competition. For fencing competitions with the round robin rounds and the elimination rounds spread out over a few days, a separate category would need a substantial number of individuals. Policy changes mainly revolve around the elite [12,48] and not the grassroots clubs or training sessions, and fencing governing bodies need to consider that, if policy changes are established throughout all fencing levels (recreational, training, and competition), this will have a large impact on the sport. Fencing training throughout all levels is often mixed, with segregation mainly during competition. If policies enforce sex segregation throughout all levels of the sport, then local fencing clubs may not survive, as it is a small niche sport [30].

## 8. Fairness in Elite Fencing: Practical Applications

Research suggests there is a 17–30% difference in sporting performance (Table 2) driven by the anthropometric and physiological differences between cis males and cis females (Table 1 and Table 3), which corresponds to fencing performance advantage for cis males over cis female fencers (Figure 1). In addition, the literature highlights that once male puberty has been experienced, testosterone suppression does not reduce all the physiological advantages, such as lean mass, strength, power, and stature, to a degree that equals cis female values. Therefore, at this time the literature suggests that there is an unfair retained physiological advantage for trans women who have experienced male puberty when participating in female fencing competitions. However, more research specifically investigating athletic transgender individuals is needed, to ascertain a clearer understanding of the long-term effect of testosterone-suppression therapy and post-puberty transitioning on physical performance across all levels of sport. A proposed solution of an open or third gender category could promote fair competition at the elite level, while allowing trans women to compete in their chosen sport. However, consultation with all groups concerned (to ensure both cis and trans women have a voice) will be needed to provide appropriate competition and inclusion for trans women fencers. It must be noted that this review focuses on the physiologically driven physical performance attributes that impact elite competitive fencing performance and it does not make recommendations for training or recreational fencing environments, such as clubs and squad training sessions, where various other control factors exist.

## Figures and Tables

**Figure 1 sports-11-00133-f001:**
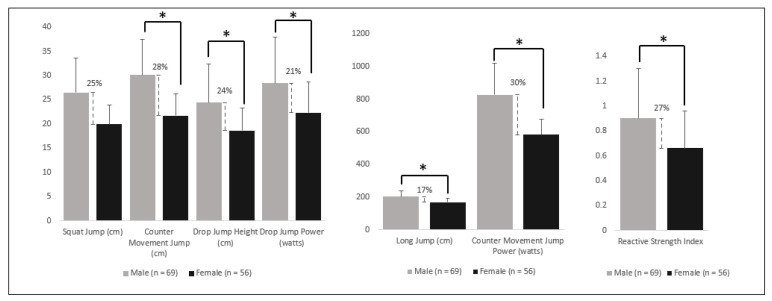
Male and female differences in fencing related fitness components adapted from Ntai et al. [37]. * = significant (*p* < 0.05), dashed line = percentage difference.

**Figure 2 sports-11-00133-f002:**
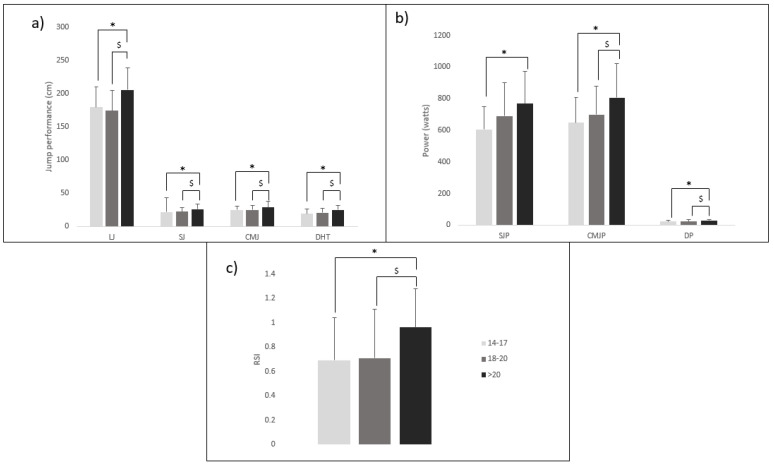
Differences in leg power between age groups of fencers adapted from Ntai et al. [78]; (**a**) top left details different jump heights for the three age groups; (**b**) top right details different jump powers for the three age groups; (**c**) bottom details reactive strength index (RSI) for the three age groups; LJ = long jump, SJ = squat jump, CMJ = counter movement jump, DHT = drop jump height, SJP = squat jump power, CMJP = counter movement jump power, DJP = drop jump power, * = 14–17 years vs. >20 years significant difference (*p* < 0.05). $ = 18 - 20 years vs. >20 years significant difference (*p* < 0.05).

**Table 1 sports-11-00133-t001:** Several anthropometric and physiological characteristics relevant to sporting performance and the corresponding approximate male advantage. Adapted from Hilton and Lundberg [18], cited in Stebbings et al. [51].

Variables	Male Advantage (%)
Limb length	~12
Cardiovascular function	~27
Muscle Mass	~37
Muscle Strength	~55

Note: First published in The Sport and Exercise Scientist, Issue 68, Summer 2021. Published by the British Association of Sport and Exercise Sciences-www.bases.org.uk.

**Table 2 sports-11-00133-t002:** Approximate male athlete percentage performance advantage in elite sport compared to female athletes. Adapted from Stebbings et al. [51] and Hilton and Lundberg [18].

Elite Sport	Male Advantage (%)
Swimming, Rowing, Running	~12
Jumping	~20
Upper body dominant	~20
Combat	≥30
Weightlifting	≥30
Throwing	≥40

**Table 3 sports-11-00133-t003:** Significant male versus female fencer differences in fencing-related anthropometric components adapted from Ntai, et al. [37]. BMI = body mass index.

	Females	Males	*p* Value
Height (cm)	167.9 ± 6.2	177.6 ± 8.9	0.001
Body mass (Kg)	57.6 ± 7.0	70.4 ± 11.9	0.001
Arm span (cm)	168.8 ± 7.2	181.8 ± 9.7	0.001
Leg length (cm)	81.7 ± 4.3	88.9 ± 6.4	0.001
BMI	20.5 ± 2.1	22.2 ± 3.0	0.001

## Data Availability

No new data were created or analyzed in this study. Data sharing is not applicable to this article.

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
