# Peer review of "The Participation of Trans Women in Competitive Fencing and Implications on Fairness: A Physiological Perspective Narrative Review"

_sports, 2023, doi:10.3390/sports11070133_

Round 1

Reviewer 1 Report

REVIEWER

This study aims to present the scientific evidence pertaining to the potentially unfair physiological advantage transgender women may have over cisgender female participants in sports and fencing in particular. The authors asserted that the evidence presented in this study can aid in the development of guidelines for the inclusion of transgender women in elite competitive women's fencing categories and provide fencing organizations with pertinent guidance. This is an intriguing, well-written, and referenced study, in my opinion. but there are a few minor flaws that must be fixed.

Introduction

Ø  Line 37 – “……throughout this review”. Do you mean the present study?

Ø  Line 62 – Please add the full name of the UCI.

Ø  Line 80 – 2 ?????

2.1. Fencing

Ø  This study focuses on the physiological aspects of transwomen's competitive fencing participation; nevertheless, the authors have limited themselves to simply energy requirements.  In my opinion, Other physiological elements of this sport should be included by the authors.

2.2. Lunge

Ø  Lines 125- 129 - Please specify the gender of participants in studies 26,31,35,43,46, and 47.

Ø  Line 151 – The authors contend that testosterone is the primary cause of gender disparities in performance. I agree, but the authors could at least mention a few more parameters that can affect sports performance, such as neuromotor control, glucose utilization, lipid metabolism, energy balance, body fat distribution, skeletal muscle fiber type, and estrogen levels.

Ø  Line 242 – Some abbreviations appear in the text for the first time. Please include the abbreviation's entire name and place it in brackets.

Ø  Line 303 – the authors stated, “However, even with these increases in muscle mass and strength, trans men are still at a disadvantage compared to cis males”. Please expand on this statement.

Ø  Lines 147-256 - The authors focused on the differences between males and women, despite the fact that transwomen are the subject of the study. Try to condense this section and, if possible, introduce the physiology and fencing performance of transgender women. If this is not feasible, this should be included as a study limitation.

Author Response

The Participation of Trans women in Competitive Fencing and Implications on Fairness: A Physiological Perspective Narrative Review

The authors would like to take this opportunity to thank the reviewers for their comments and have to the best of their ability adjusted the manuscript appropriately. Please find below an outline of the changes made to the manuscript and discussion around some points raised.

Reviewer:  Line 37 – “……throughout this review”. Do you mean the present study?

Response: Line 37 - ‘indicated throughout this review’ changed to ‘abbreviated to ‘trans’ throughout our review’

Reviewer: Please add the full name of the UCI.

Response: Apologies, the full name of ‘Union Cycliste Internationale’ added before the abbreviated UCI

Reviewer: Line 80 – 2 ?????

Response: We have renumbered the sections, so hopefully this reads better.

 Reviewer:  This study focuses on the physiological aspects of transwomen's competitive fencing participation; nevertheless, the authors have limited themselves to simply energy requirements.  In my opinion, Other physiological elements of this sport should be included by the authors.

Response:  We have structured the intro and included fencing. We have made the specific amendments below and hope this highlights the main elements of fencing. Ideally it is a needs analysis of the main physiological elements which is presented in the literature.

  • Lines 73-123 - the fencing and lunge section (originally section 2) has been incorporated into the introduction as part of a needs analysis of fencing.
  • Lines 73 - 100 - what was the section 2. Fencing is now Section 1.3.1 Physiological Demands of Elite Fencing - there is limited research specific to Elite Fencing and therefore we have included the main findings that we could find and discuss in relation to gender differences.

Reviewer: Lines 125- 129 - Please specify the gender of participants in studies 26,31,35,43,46, and 47.

Response: The gender of participants is not stated in this section as this is an overview of Fencing as a sport and the demands of the athletes participating in this sport. The gender specific differences are discussed later in sections 2 & 3.

Reviewer: Line 151 – The authors contend that testosterone is the primary cause of gender disparities in performance. I agree, but the authors could at least mention a few more parameters that can affect sports performance, such as neuromotor control, glucose utilization, lipid metabolism, energy balance, body fat distribution, skeletal muscle fiber type, and estrogen levels.

Response: We have added acknowledgement that testosterone also affecting many other bodily mechanisms but have for this review focused on the anthropometric and physiological parameters within table 1 and fencing specific variable identified during section 2 & 3

Reviewer: Line 242 – Some abbreviations appear in the text for the first time. Please include the abbreviation's entire name and place it in brackets.

Response: Apologies, we have gone through and added the full name when first mentioned.

Reviewer: Line 303 – the authors stated, “However, even with these increases in muscle mass and strength, trans men are still at a disadvantage compared to cis males”. Please expand on this statement.

Response: Thank you for the comment, we have now expanded on this and added data that exists comparing cisgender and trans men to support this comment. 

Reviewer: Lines 147-256 - The authors focused on the differences between males and women, despite the fact that transwomen are the subject of the study. Try to condense this section and, if possible, introduce the physiology and fencing performance of transgender women. If this is not feasible, this should be included as a study limitation.

Response: The discussion regarding the difference between cis males and cis females was included as we believe it is an essential branch of evidence due to a severe lack of research involving trans participants and none to our knowledge investigating elite trans athletes. Therefore, the inclusion and focus on comparable populations such as cis male vs cis female, those with genetic variations, and testosterone suppressed cis males where used to infer current evidence based advise. We have tried to highlight this fact throughout the review and that more research and continual re-evaluation of new evidence is needed to correctly and fairly enact any policy or exclusion criteria changes for specific sporting governing bodies.

Reviewer 2 Report

Dear authors,

Thank you for your paper.

Some concerns about this review:

- Delete green color from keywords 

- Line 27 delete a double blank

-  I consider it would be important include a section with the different factors that affect to Trans women (hormonal or physiological aspects, not only testosterone). Similar to section 3.1 but briefly

- Include title section in 2. (Line 80). If not, include this sections in introduction. But I consider important to include some aspects about: how did you the review; what kind of study design is this? On the other hand a section with the procedure to review the literature would be necessary. It could be interesting if you could include the quality criterial you have used to this review

- in the studies you include, Are any study considered the statistical difference are not only % difference? Are there some study where they compare the physiological values and performance with men and trans women? (Or men and women only if not. In order to see the physiological and performance results differences. Like study from table 4 but with statistical are not only descriptive values)

- line 398: delete blank

- review reference section following the Sport journal rules (for instance black computer in journal publication year)

Congratulations for your work.

Author Response

The authors would like to take this opportunity to thank the reviewers for their comments and have to the best of their ability adjusted the manuscript appropriately. Please find below an outline of the changes made to the manuscript and discussion around some points raised.

Reviewer: Delete green color from keywords

Response:  This has been removed.

Reviewer: Line 27 delete a double blank

Response: Apologies, the double blank removed

Reviewer: I consider it would be important include a section with the different factors that affect to Trans women (hormonal or physiological aspects, not only testosterone). Similar to section 3.1 but briefly

Response: This review has focused on the what the authors considered the most influential hormonal difference between cis male and cis females, testosterone.  There is a lack of research currently available involving elite sports or trained transgender participants hence our reliance on comparable populations as outlined previously to make inferences to aid in policy and inclusion criteria decisions.

Reviewer: Include title section in 2. (Line 80). If not, include this sections in introduction. But I consider important to include some aspects about: how did you the review; what kind of study design is this? On the other hand a section with the procedure to review the literature would be necessary. It could be interesting if you could include the quality criterial you have used to this review

Response: Thank you for your comment. We have restructured the beginning slightly so that the section Fencing is now Section 1.3. Needs Analysis of Elite Fencing, split into 2 sections 1.3.1 Physiological demands of Elite Fencing and 1.3.2 Lunge in Elite Fencing, all of which are now included within the Introduction section. As this was a narrative review, we have not used a specific strategy as you would do for a systematic literature review. Therefore, we have not included a methodological statement. There is a section 1.4 which touches on the methodological considerations which discusses the quality and limitations of some of the literature.

Reviewer: in the studies you include, Are any study considered the statistical difference are not only % difference? Are there some study where they compare the physiological values and performance with men and trans women? (Or men and women only if not. In order to see the physiological and performance results differences. Like study from table 4 but with statistical are not only descriptive values)

Response:  Unfortunately the studies relating to fencing only presented the descriptive differences however, we have attempted to include the statistical differences when available.

Reviewer: - line 398: delete blank

Response: Apologies, we have deleted the double blank.

Reviewer: review reference section following the Sport journal rules (for instance black computer in journal publication year)

Response: We have gone back through and carefully checked the reference section.